# Genetic and clinical determinants of neonatal jaundice and growth patterns in the Qingdao birth cohort: A genome-wide association study

Xu Chen[1,2], Peina Du[3,4], Shuo Li[5], Xiaohong Wang[2], Mengyang Xu[2], Zhaobin Chu[2], Yue Zhang[2], Zhengyang Yao[1], Xuejie Huan[1], Yushan Huang[6], Mingyan Fang[6], Ya Gao[6], Guangyi Fan[2], Xin Jin[6]*, Hui Huang[4]*, Silin Pan[5]*

1 Department of Physiology, Shandong Provincial Key Laboratory of Pathogenesis and Prevention of Neurological Disorders and State Key Disciplines: Physiology, School of Basic Medicine, Qingdao University, Qingdao, China, 2 BGI Research, Qingdao, China, 3 Clin Lab, BGI Genomics, Qingdao, China, 4 BGI Genomics, Shenzhen, China, 5 Qingdao Women and Children's Hospital, Qingdao University, Qingdao, China, 6 BGI Research, Shenzhen, China

☙ These authors contributed equally to this work.
* jinxin@genomics.cn (XJ); huanghui@genomics.cn (HH); silinpan@126.com (SP)

## Abstract

### Background

Neonatal jaundice and early growth patterns are important indicators of early health, and genetic as well as clinical factors are known to influence these traits. However, evidence from East Asian newborns is still limited.

### Results

This study presents a genome-wide association study (GWAS) investigating genetic determinants of Healthy Newborn Growth Indicators (HNGI), with a focus on neonatal jaundice (JAU), jaundice resolution (JAUR), birth weight (BW), and growth metrics (weight, height, BMI) measured within 90–105 days after birth. Our analysis identified 778 single-nucleotide polymorphisms (SNPs) across 120 genes significantly associated with HNGI. Among these associated variants, we found 12 missense mutations, seven of which are novel. The most significant association signal was for rs4148323 ($P = 2.3 \times 10^{-18}$) within the *UGT1A* gene locus, a well-established variant for JAU. Additionally, we discovered three novel missense mutations associated with JAU and JAUR. For BW, a novel missense mutation, rs148399850 ($P = 2.3 \times 10^{-8}$), was identified in the *ATP7* gene, suggesting *ATP7* as a potential functional regulator of birth weight. Analysis of allele frequency distributions across global and Chinese populations revealed distinct patterns of genetic diversity. Functional enrichment analysis of the candidate genes highlighted 18 genes significantly involved in 14 essential metabolic pathways related to growth and development. Furthermore, an analysis of clinical risk factors using data from the Biobank Japan (BBJ) demonstrated significant influences of various clinical conditions on HNGI.

**Data availability statement:** The data supporting the findings of this study have been deposited in the CNGB Sequence Archive (CNSA) of the China National GeneBank Database (CNGBdb) and are accessible at: https://db.cngb.org/data_resources/project/CNP0005415.

**Funding:** The author(s) received no specific funding for this work.

**Competing interests:** The authors have declared that no competing interests exist.

**Abbreviations:** AFR, Africans; AHR, Aryl Hydrocarbon Receptor; AMR, Americans; ARF, Acute renal failure; BBJ, Biobank Japan; BMPR1B, Bone Morphogenetic Protein Receptor Type 1B; BTAF1, B-TFIID TATA-Box Binding Protein Associated Factor 1; BW, Birth weight; Car, Cardiomegaly; CCDC65, Coiled-Coil Domain Containing 65; CHF, Chronic heart failure; Cir, Cirrhosis; CRF, Chronic renal failure; CTSZ, Cathepsin Z; DLG2, Disks large homolog 2; EAS, East Asians; EUR, Europeans; Go, Gene ontology; GWAS, Genome-wide association analysis; HC, Hypertrophic cardiomyopathy; HDLC, HDL-cholesterol; HNGI, Healthy Newborn Growth Indicators; HSD17B3, Hydroxysteroid 17-Beta Dehydrogenase 3; JAU, Jaundice; JAUR, Jaundice resolution; KCNMA1, Potassium Calcium-Activated Channel Subfamily M Alpha 1; MCH, Mean corpuscular hemoglobin; MCHC, Mean corpuscular hemoglobin concentration; MHC, Major histocompatibility complex; NS, Nephrotic syndrome; PPP1CB, Protein Phosphatase 1 Catalytic Subunit Beta; PRKCH, Protein Kinase C Eta; PRKG1, Protein Kinase CGMP-Dependent 1; SAS, South Asians; SCS, Spinal canal stenosis (SCS); SNPs, Single nucleotide polymorphisms; TMEM236, Transmembrane Protein 236; WNT10B, Wnt Family Member 10 B.

## Conclusions

This comprehensive analysis of newborns from the Qingdao cohort provides critical data for expanding molecular marker databases for prenatal screening, offering early warnings for jaundice, and guiding the healthy growth of newborns.

## Introduction

The early stages of human development are critically important, with the newborn period playing a pivotal role in determining long-term health outcomes [1]. This phase is marked by profound physiological changes and considerable vulnerability, necessitating a comprehensive understanding of the factors that promote healthy growth [2]. While traditional health assessments have largely focused on adult metrics, diseases, and lifestyle-related risk factors [3–5], comprehensive health evaluations in newborn cohorts remain relatively rare.

Neonatal jaundice represents one of the most common medical conditions in the first two weeks of life, affecting approximately 60% of full-term and 80% of preterm infants within the first week after birth [6]. Although most cases are mild and self-limiting, failure to identify pathological forms can lead to severe complications, including bilirubin-induced neurologic injury. Established risk factors include prematurity, enzymatic immaturity (e.g., reduced *UGT1A1* activity), hemolytic disease, and breastfeeding-related issues. Given its high incidence and potential adverse outcomes, elucidating the genetic and clinical determinants of neonatal jaundice is essential for improving early-life health.

Advances in genomic research, particularly genome-wide association studies (GWAS), have revolutionized our understanding of the genetic basis of various health conditions [7–9]. These studies provide valuable insights into the genetic factors influencing neonatal growth indicators, shedding light on the biological mechanisms governing early development [10]. For instance, prior research has implicated the cathepsin Z (CTSZ) locus in jaundice progression [11] and highlighted significant correlations between UGT1A gene variants and jaundice susceptibility [12]. In the context of growth metrics, GWAS have identified key genes such as FTO (fat mass and obesity-associated) [13], and a meta-analysis of birth weight in 86,577 women revealed five novel loci [14]. Nevertheless, neonatal growth is a complex trait shaped by numerous genetic, environmental, and developmental factors [15]. Traditional GWAS approaches are often limited by methodological constraints [16] and insufficient account of genetic diversity across populations, complicating the interpretation of neonatal health outcomes. This underscores the need for more inclusive analytical frameworks that incorporate population-specific genetic information and integrated tools to enhance discovery for complex traits [17,18].

Examining allele frequency distributions across global populations can clarify the differential effects of genetic variants, while stratified analyses within specific regions, such as diverse Chinese cohorts—may reveal important intra-national variations [19]. Functional analyses using pathway databases such as KEGG further

contribute to understanding the molecular mechanisms underpinning healthy development [20]. Additionally, maternal risk factors, a critical consideration in neonatal studies—have been increasingly characterized through large-scale biobank initiatives. For example, the Biobank Japan (BBJ) project has identified associations between vasospastic angina and fatal myocardial infarction risk [21], and demonstrated that BMI stratification can refine polygenic predictions of type 2 diabetes [22]. Integrating maternal risk profiles with genetic data offers a more holistic perspective on neonatal health [23].

The application of comprehensive GWAS in neonatal research has been transformative, providing new insights into the genetic architecture of early-life conditions [24]. Combining GWAS findings with clinical risk assessments not only helps quantify individual disease susceptibility but also facilitates validation of genetic signals [25]. In summary, this study aims to advance the field of neonatal health by addressing existing gaps in comprehensive genetic analyses. Through the integration of diverse datasets and consideration of population-specific genetic factors, we seek to contribute to improved neonatal care strategies and pave the way for personalized medical interventions in early life.

## Methods

### Sample collections and data processing

This study was based on the Qingdao West Coast New Area newborn whole-genome sequencing (WGS) program [26]. A total of 9,992 newborns were randomly recruited from Jan 2021 to Oct 2022, of which 7,140 newborns' parents agreed to participant the research. After data processing and quality control in another unpublished research, 6,685 newborns were used for further analysis in our study. To reduce the influence of developmental immaturity and medical intervention, cases identified as preterm or those who had recently received blood transfusions were excluded from the analysis.

We collected newborns BW through questionnaires. Other birth traits such as JAU, JAUR, height, weight and BMI at 90–105 days were obtained from the Maternal and Child Health Handbook in Qingdao city.

The study protocol was approved by the Research Ethics Committee of Qingdao Women and Children's Hospital (approval no. QFELL-KY-2020-29) and the Institutional Review Board of Bioethics and Biosafety at BGI (BGI-IRB; approval no. BGI-IRB 20064). Written informed consent was provided by the guardians of all participating newborns. Clinical trial number: (2020)CJ1059.

### Genome-wide association analysis

Before association analysis, a series of quality control measures were performed: (1) HWE > 0.000001. (2) genotyping rates ≥ 90%. (3) minor allele frequencies (MAF) ≥ 1%. (4) Principal component analysis (PCA) was used to correct for population stratification, with EIGENSOFT [27] software used for this analysis. Relatedness between samples was determined using Identical by Descent (IBD) analysis with PLINK (v1.9) [28]. Unrelated newborns were identified using the proportion of relative of PI_HAT < 0.1875. Ultimately, 6,631 unrelated individuals were determined and 6,515,339 SNPs passed quality control filters and were included in subsequent analysis. We used plink v1.9 to detect the association between genotypes and phenotypes, followed by --linear for quantitative traits and --assoc for binary traits. For quantitative traits, we excluded samples with missing values and values beyond 4 s.d. from the mean. Gender and the first five principal components were applied as covariates. The GWAS catalog database [29] (e107_r2022-08-26) was used to define known and novel loci. The LD score regression approach [30] was used to estimate the SNP heritability. Locuszoom [31] was applied to visualize the loci. Expression-associated SNPs were derived from the GTEx v8 database (https://www.gtexportal.org/home/).

We further compared the allele frequency of missense variants ($P < 1 \times 10^{-5}$) between our cohort and three large population cohorts, including five subpopulations of 1000 Genome phase3 dataset, ChinaMAP, and gnomAD [31–33]. The provincial allele frequencies were obtained from the Chinese Millionome database [3].

## KEGG pathway and Gene Ontology (GO) analysis

We conducted KEGG pathway and Gene Ontology (GO) analyses on candidate genes from the GWAS using the online tool DAVID (https://david.ncifcrf.gov). *P*-value < .05 was considered to be statistically significant.

## Genetic correlation estimation analysis

We used LDSC to estimate the genetic correlation between each trait and clinical factors from BBJ. SNPs within the major histocompatibility complex (MHC) region (chr6:25-34Mb) were excluded from the analysis due to its complex LD structure. The b37 coordinates were converted to b38 coordinates using the liftover script from UCSC.

The leave-one-out method [34] was used to evaluate the sensitivity of the selected SNPs (IVs).

## Results

### Indicators of Healthy Newborn Growth Guidelines (HNGI)

To evaluate neonatal growth and provide health guidance, our study collected data on neonatal JAU occurrence and JAUR, BW, and growth metrics (height, weight, BMI) between 90–105 days, which are not negligible HNGI. A GWAS was conducted on HNGI, aiming to identify genetic factors influencing newborn health. In total, we found 778 significantly associated SNPs ($P < 1 \times 10^{-5}$), corresponding to 120 genes. Among them were 12 missense mutations, containing 7 novel missense mutations (Fig 1 and S1 Fig). Leave-one-out results showed that the MR estimate did not change considerably after eliminating SNPs one by one (S5 Fig). In silico functional annotation using SIFT and PolyPhen-2 showed mixed predictions among these seven missense variants, with most predicted to be tolerated or benign, while a few (e.g., rs1042597 and rs149243426) were suggested to have potential functional impact based on SIFT scores (S12 Table).

As anticipated, genomic region analyses revealed significant associations between the *UGT1A* gene cluster on chromosome 2 and both neonatal jaundice (JAU) and jaundice resolution (JAUR) (Fig 2A and 2B). GWAS identified 80 SNPs significantly associated with JAU and 70 with JAUR ($P < 5 \times 10^{-8}$) (S1 and S2 Tables, S2 Fig).

Since missense mutations frequently alter protein function, we prioritized all significantly associated missense SNPs ($P < 5 \times 10^{-8}$). Among these, eight missense variants—rs1042597, rs17868323, rs17868324, rs11692021, rs6759892, rs2070959, rs1105879, and rs4148323—were significantly associated with both JAU and JAUR. Of these, three SNPs (rs1042597, rs17868323, and rs17868324) are novel and may represent valuable targets for further investigation into the genetics of neonatal jaundice.

The strongest association signal for JAUR was observed for rs4148323 ($P = 2.3 \times 10^{-18}$), a missense mutation located in the first exon of the *UGT1A1* gene. This variant has been widely reported in Asian populations, including Chinese cohorts, as a key determinant of serum bilirubin levels [35].

The replication of this well-established association underscores the reliability of our GWAS findings and provides a validated reference point for elucidating the molecular mechanisms underlying neonatal jaundice.

Subsequently, we analyzed the GWAS results of BW, and found that 9 significantly associated SNPs ($P < 5 \times 10^{-8}$) (Fig 2C and S3 Table), of which, a novel SNP rs148399850 ($P = 2.3 \times 10^{-8}$) is a missense mutation in the *ATP7b* gene, and previous studies have reported weight loss in *atp7b* knockout mice [36]. At Bayesian thresholds, 11 SNPs were significantly associated with height (Fig 2D and S4 Table), 6 SNPs were significantly associated with weight (Fig 2E and S5 Table), and no significantly associated SNPs were found in BMI (Fig 2F and S6 Table). This was apparently excessive Bayesian threshold leading to false negatives, and we adjusted the cut off to $P = 1 \times 10^{-5}$. Subsequently, we counted 70 suggestive associations in the height results, of which rs927472313($P = 1.1 \times 10^{-6}$), a missense mutation, is located in the *TMEM236* (Transmembrane Protein 236) gene (Fig 2E). In addition, we found 231 suggestive associations in weight ($P < 1 \times 10^{-5}$) (S5 Table), on which, two missense mutations, rs149243426 ($P = 4.0 \times 10^{-6}$) and rs10747556 ($P = 4.1 \times 10^{-6}$), located in the *BTAF1* (B-TFIID TATA-Box Binding Protein Associated Factor 1) and *CCDC65* (Coiled-Coil Domain Containing 65) genes,

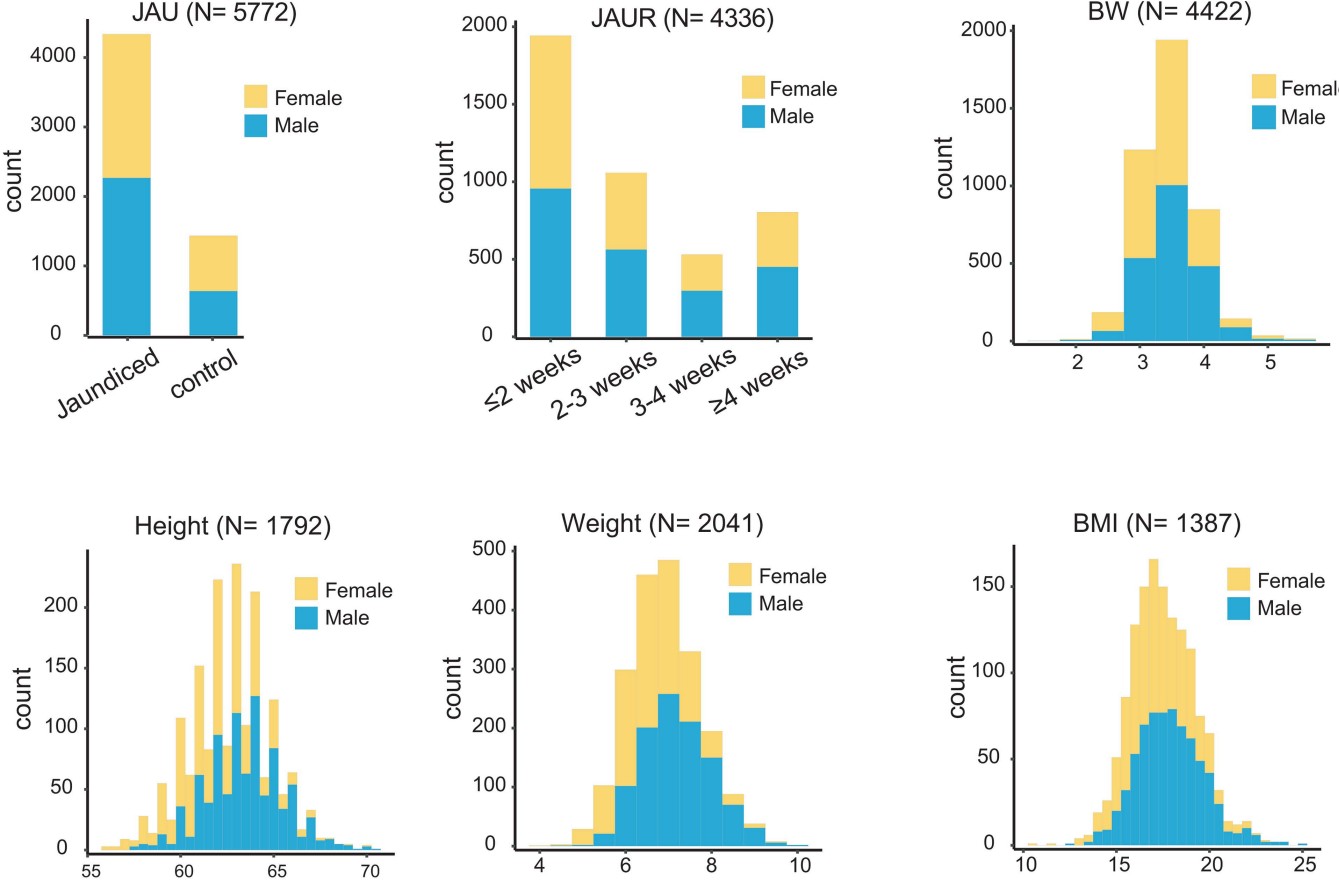

**Fig 1. Distribution of the phenotypes across the newborn growth indicators.** JAU: jaundice. JAUR: jaundice regression. BW: birth weight. BMI: body mass index.

respectively. In terms of BMI, we counted 109 suggestive associations ($P < 1 \times 10^{-5}$), whereas no missense mutation SNPs were found (S6 Table).

## Allele frequency distribution of missense mutation SNP

Given the importance of missense mutations in HNGI, we compared the allele frequencies of the 12 identified missense mutations across the ChinaMAP dataset, the gnomAD database, and five super-populations from the 1000 Genomes Project (East Asians [EAS], South Asians [SAS], Africans [AFR], Europeans [EUR], and Admixed Americans [AMR]) (Fig 3A).

Distinct allele frequency patterns were observed for missense mutations associated with JAU and JAUR. The alleles of rs17868323, rs17868324, rs6759892, and rs1105879 were more frequent in gnomAD, SAS, AFR, EUR, and AMR populations compared to the Qingdao cohort, ChinaMAP, and EAS. In contrast, the opposite frequency pattern was evident for rs1042597 and rs4148323 (S3 Fig). The alleles of rs11692021 and rs2070959 were more prevalent in EUR, SAS, and the aggregate gnomAD dataset than in other groups.

For missense mutations associated with growth metrics, rs10747556 (Weight) exhibited considerable diversity in allele frequency across different cohorts (S7 Table). Meanwhile, the distributions of rs148399850 (BW) and rs927472313 (Height) showed little differentiation across the populations studied.

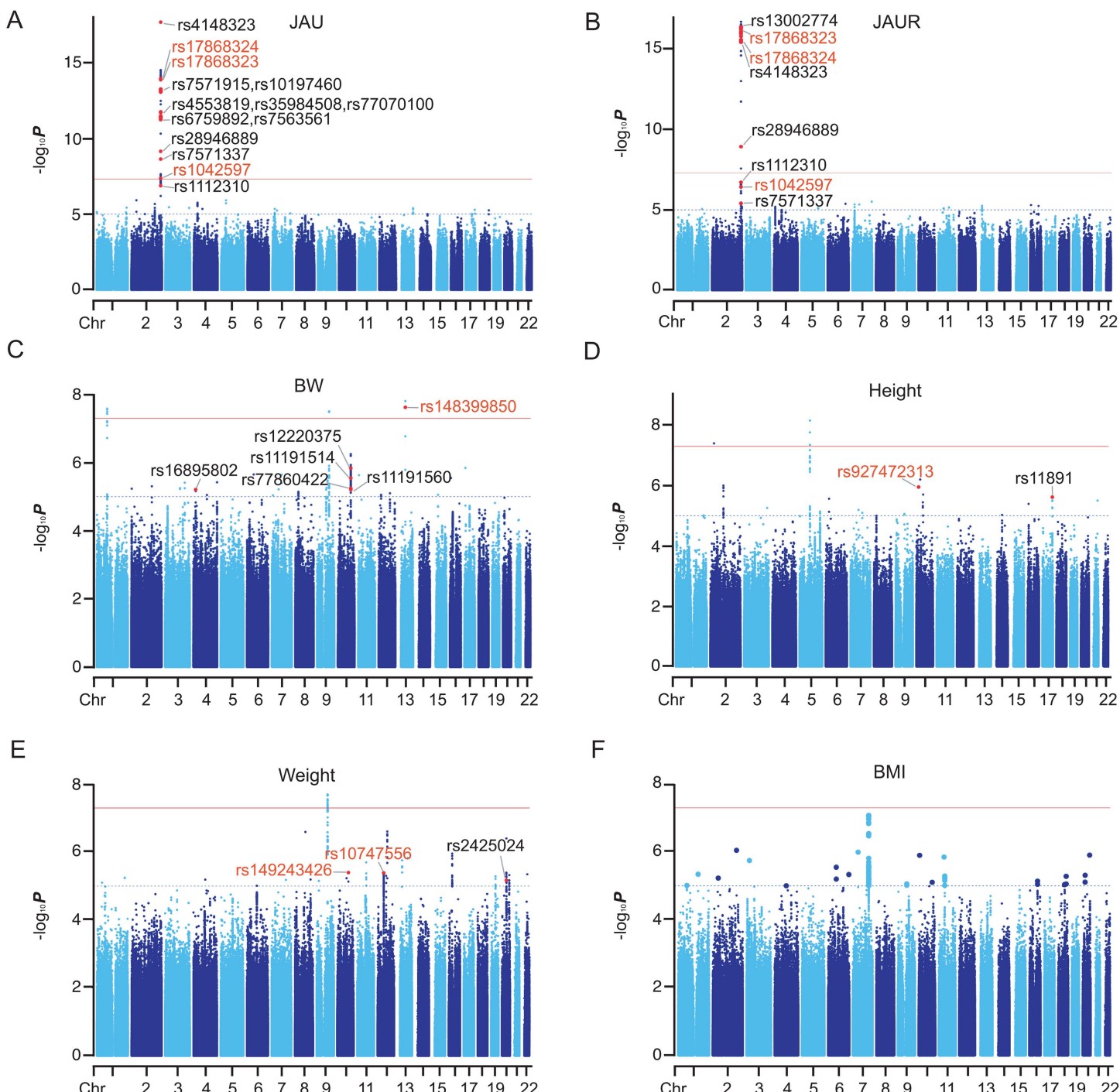

**Fig 2. Manhattan plots for GWAS analysis on JAU, JAUR, BW, Height, Weight and BMI.** The blue line represents the threshold ($-\log_{10}(1 \times 10^{-5})$) for the GWAS statistical significance. Redline is $-\log_{10}(5 \times 10^{-8})$. Known SNPs, defined as significant with the investigated trait in the GWAS catalog (e107_r2022-08-26) are marked in black (For space reasons, some SNPs in JAU and JAUR are not shown by name, please see S1 and S2 Tables for all SNPs). Novel missense SNPs are marked in red.

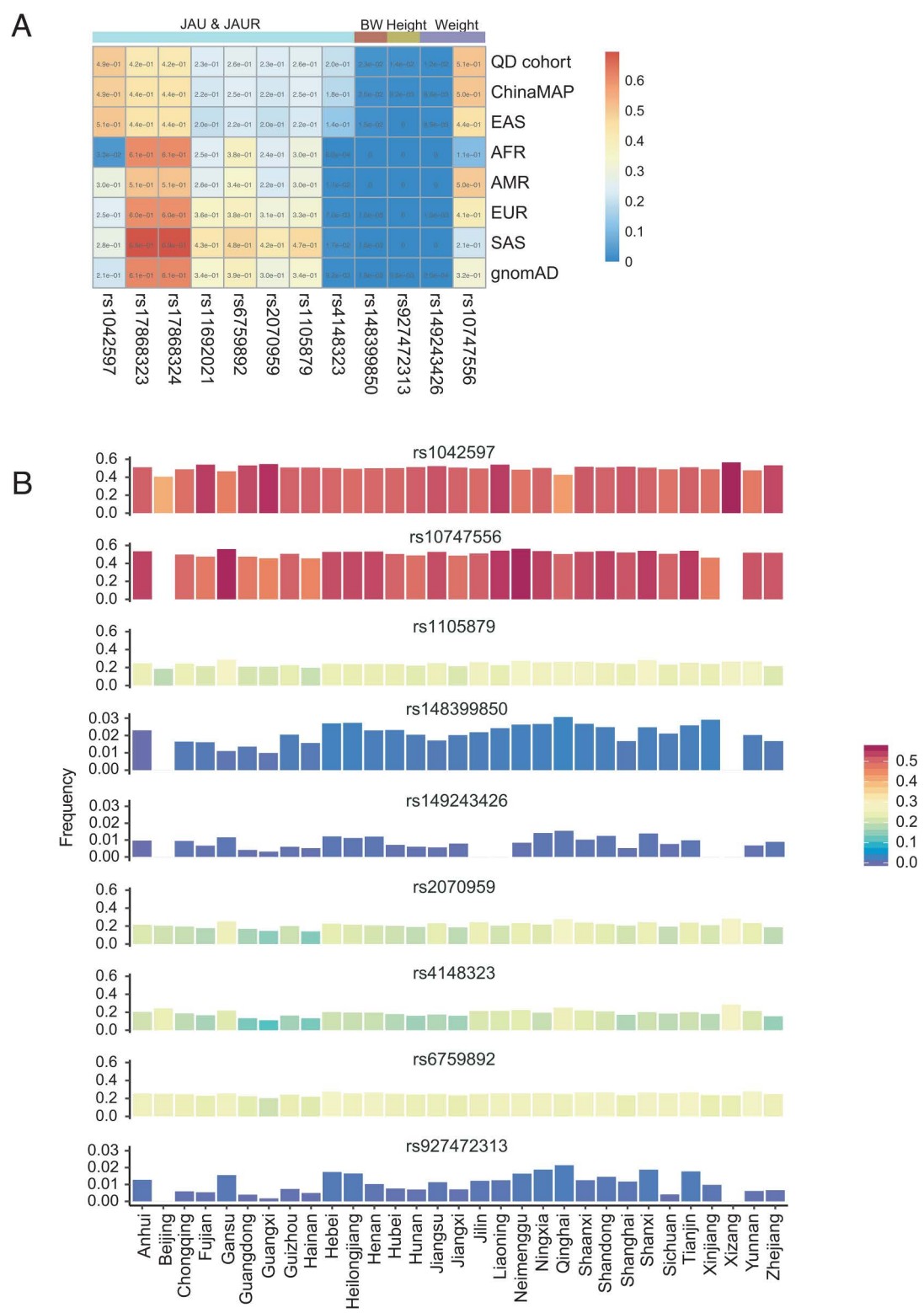

**Fig 3. Landscape of missense variant allele frequencies across global populations and within China. (A)** Comparison of allele frequency of missense variants ($P < 1 \times 10^{-5}$) between our cohort(QD cohort) and other populations including ChinaMAP, gnomAD and 1000 Genome Project dataset (East Asians (EAS), South Asians (SAS), Africans (AFR), Europeans (EUR) and Americans (AMR)). **(B)** Allele frequency per Chinese administrative division for missense variants ($P < 1 \times 10^{-5}$) based on the Chinese Millionome Database (CMDB).

Since we are using a QD cohort, we further analysed 9 missense mutations in the Chinese Millionome Database (3 missense mutations not in the database) (Fig 3B and S8 Table), and found that rs1042597, rs2070959, rs4148323, and rs10747556 had the highest allele frequency in Xizang Province, but rs148399850, rs927472313 and rs149243426 were undistributed in Xizang and Beijing. rs6759892, rs2070959, rs4148323 and rs10747556 were almost undistributed in Guangxi Province. Regarding rs1105879, we only observed a lower allele frequency on the southeast coast. In terms of the overall trend, rs148399850 and rs927472313 seem to show a decrease in allele frequency from northern to southern regions. It is worth noting that rs927472313 is a missense mutation in height, and the height trend of the Chinese population shows a decrease from north to south.

Furthermore, $\chi^2$ tests across 31 provincial-level populations revealed significant heterogeneity for six variants (rs1042597, rs6759892, rs2070959, rs1105879, rs4148323, and rs10747556; $P \le 0.0045$), confirming the population-specific patterns observed in S8 Table.

## Function enrichment analysis HNGI genes

Subsequently, we performed a comprehensive examination of 120 candidate genes from the HNGI (Fig 4A and S4 Fig), revealing 20 arrays of gene ontology (GO) (S9 Table) and 18 candidate genes (includes 9 genes in the *UGT1A* family) significant enrichment to NHGI. In the results of GO, the most significant GO we found was GO:0052696: flavonoid glucuronidation ($P = 1.1 \times 10^{-18}$), which is extremely relevant to growth and development. The remaining GO arrays also inform the understanding of the molecular network of the NHGI. And the results of the genes, we found that *HSD17B3* (Hydroxysteroid 17-Beta Dehydrogenase 3) in BW, *PPP1CB* (Protein Phosphatase 1 Catalytic Subunit Beta), *PRKCH* (Protein Kinase C Eta), and *PRKG1* (Protein Kinase CGMP-Dependent 1) in Height, *AHR* (Aryl Hydrocarbon Receptor) and *UGT1A* family genes in JAU, *DLG2* (Disks large homolog 2) and *UGT1A* family genes in JAUR, as well as *BMPR1B* (Bone Morphogenetic Protein Receptor Type 1B), *KCNMA1* (Potassium Calcium-Activated Channel Subfamily M Alpha 1), and *WNT10B* (Wnt Family Member 10 B) in Weight.

Further analysis of KEGG pathways identified 14 metabolic pathways (S10 Table), with the hsa00053, Ascorbate and Aldarate metabolism pathway being the most significantly enriched ($P = 8.8 \times 10^{-13}$). Additionally, the presence of enriched pathways crucial for growth and development, such as hsa00040, Pentose and glucuronate interconversions ($P = 4.4 \times 10^{-12}$), hsa00140, Steroid hormone biosynthesis ($P = 1.1 \times 10^{-11}$), hsa00830, Retinol metabolism ($P = 9.7 \times 10^{-10}$), hsa00980, Metabolism of xenobiotics by cytochrome P450 ($P = 3.0 \times 10^{-10}$), hsa01240, Biosynthesis of cofactors ($P = 6.0 \times 10^{-7}$), hsa04270, Vascular smooth muscle contraction ($P = .027$), and hsa04390, Hippo signaling pathway ($P = .041$), underscores the importance of the HNGI as a guideline for promoting healthy newborn growth. Moreover, our results also revealed two pathways, hsa00860, Porphyrin metabolism ($P = 2.1 \times 10^{-11}$) and hsa04976, Bile secretion ($P = 8.6 \times 10^{-9}$) which are tightly associated with JAU and JAUR. We also identified two drug-associated pathways, hsa00982, Drug metabolism -cytochrome P450 ($P = 1.5 \times 10^{-9}$) and hsa00983, Drug metabolism-other enzymes ($P = 3.6 \times 10^{-9}$), as well as two chemical carcinogenesis-associated pathways, hsa05204, Chemical carcinogenesis-DNA adducts ($P = 1.2 \times 10^{-9}$) and hsa05207, Chemical carcinogenesis – receptor activation ($P = 6.3 \times 10^{-7}$), and the above findings provide a certain molecular reference for us to improve neonatal drug guidance. In terms of gene distribution, it can measure the coverage of molecular markers, and our results show that except for chromosome 15, the remaining 21 chromosomes were distributed (Fig 4B).

## Causal effects of clinical factors on indicators of HNGI

To assess the genetic correlation between the clinical factors and HNGI, we used 54 risk factors from the BBJ database and we found multiple sets of apparent correlations ($|rg| \ge 0.45$) (Fig 5A and S11 Table). We found that Hypertrophic cardiomyopathy (HC) positively correlated with BW (rg = 0.88), while Spinal canal stenosis (SCS) (rg = −0.59), HDL-cholesterol (HDLC) (rg = −0.55), Mean corpuscular hemoglobin concentration (MCHC) (rg = −0.45), and Calcium (Ca)

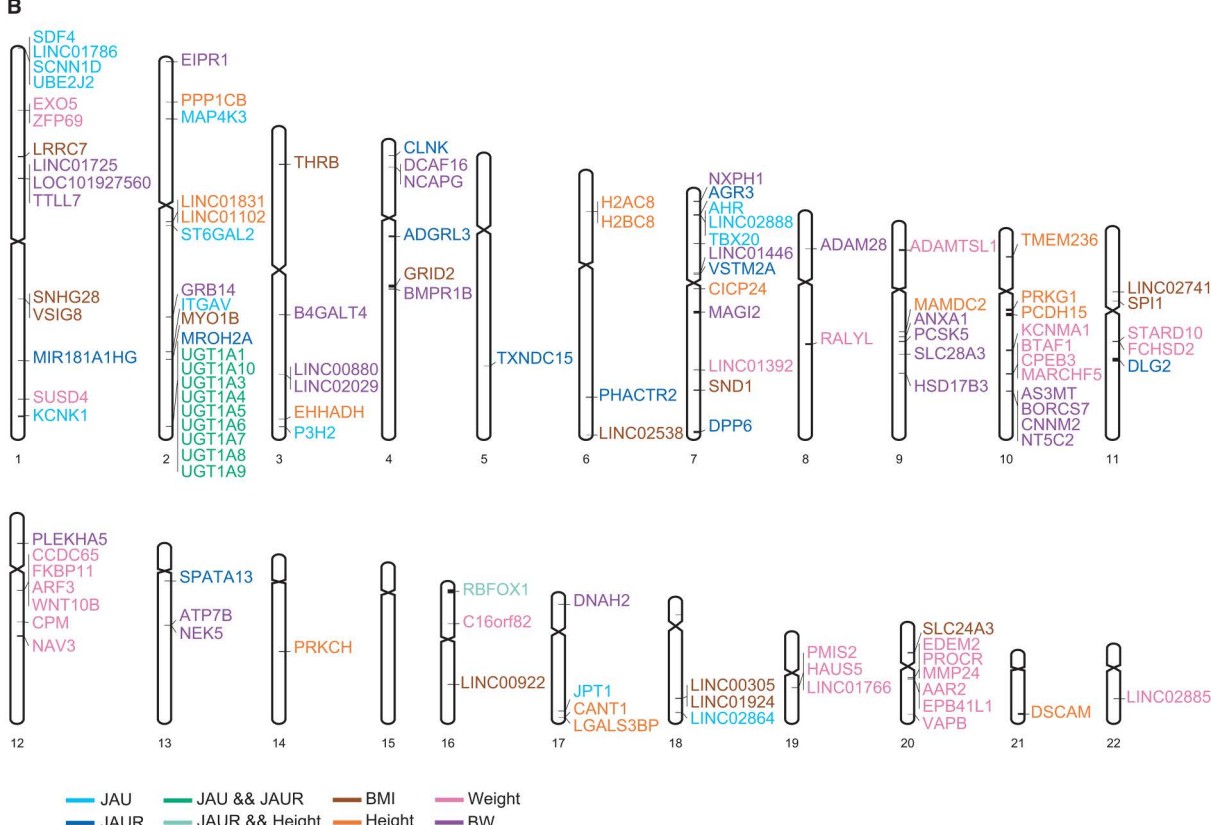

**Fig 4. Functional analysis of all GWAS candidate genes and distribution of candidate genes on chromosomes. (A)** KEGG pathway enrichment analysis of all GWAS candidate genes. $P$- < .05 was considered to be statistically significant. hsa00053, ascorbate and aldarate metabolism; hsa00040, pentose and glucuronate interconversions; hsa00140, steroid hormone biosynthesis; hsa00860, porphyrin metabolism; hsa00830, retinol metabolism;

hsa05204, chemical carcinogenesis-DNA adducts; hsa00982, drug metabolism-cytochrome P450; hsa00980, metabolism of xenobiotics by cytochrome P450; hsa00983, drug metabolism-other enzymes; hsa04976, bile secretion; hsa01240, biosynthesis of cofactors; hsa05207, chemical carcinogenesis-receptor activation; hsa04270, vascular smooth muscle contraction; hsa04390, hippo signaling pathway. **(B)** The distribution of candidate genes for GWAS on chromosomes. Different colors represent different traits.

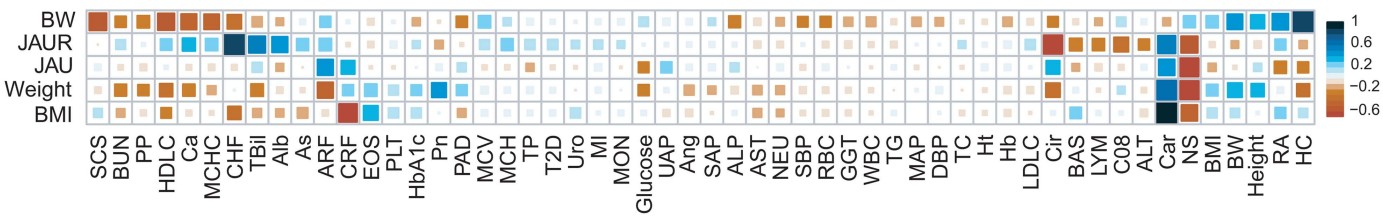

**Fig 5. Genetic correlation analysis of newborn growth indicators with clinical factors.** Genetic correlation for clinical factors in BBJ and JAU, JAUR, BW, weight and BMI estimated by LDSC. The genetic correlation estimates ($r_g$) are colored according to their intensity and direction.

(rg = −0.48) negative correlation with BW. Chronic heart failure (CHF) positively correlated with JAUR (rg = 0.86). Cirrhosis (Cir) negative correlation with JAUR (rg = −0.69). Acute renal failure (ARF) positively correlated with JAU (rg = −0.49), but negative correlation with weight (rg = −0.49). Chronic renal failure (CRF) negative correlation with BMI (rg = −0.65). Cardiomegaly (Car) positively correlated with JAU, JAUR, Weight, and BMI (rg = 0.50, 0.61, 0.72, and 1.03), while Nephrotic syndrome (NS) negative correlation with them (rg = −0.67, −0.55, −0.68, and −0.50).

## Discussion

The intricate relationship between genetics and clinical conditions in shaping neonatal growth and development represents a burgeoning field of inquiry with profound implications for early-stage healthcare. Our study leverages HNGI GWAS to elucidate the genetic underpinnings of healthy newborn growth, identifying 778 significantly SNPs across 120 genes. This vast array of genetic markers underscores not only the polygenic nature of neonatal development but also the nuanced interplay between genetics and clinical outcomes, highlighting novel insights and potential avenues for clinical application. The *UGT1A* gene clusters significant association with JAU and JAUR underscores its pivotal role in bilirubin metabolism and neonatal health. This aligns with previous findings [12], which also pinpointed *UGT1A* involvement in bilirubin processing disorders. Furthermore, the strongest association with JAUR was found for rs4148323 [35], has been commonly reported to be associated with serum bilirubin levels in Asian populations, including Chinese. This provides further evidence of the reliability of the NHGI GWAS. Given the crucial role missense mutations play in genetic variation, 3 novel missense SNPs of JAU and JAUR—rs1042597, rs17868323, and rs17868324 contributes to the expanding genetic landscape associated with neonatal conditions. Subsequently, our GWAS results for BW identified a novel SNP rs148399850, a missense mutation in the *ATP7b* gene. Since *ATP7b* knockout mice had significantly lower body weights [36], we hypothesized that the *ATP7b* gene may have a similar role in neonates, and also that rs148399850 may be an important molecular marker for the regulation of BW. Then, we are focusing strongly on missense mutations, and we found rs927472313 in the GWAS results for height, is located in the *TMEM236* gene. Past studies have understood the *TMEM236* highly expressed in pancreatic α cells that secrete glucagon, elevating blood glucose, also considered to be a potential novel diagnostic biomarker for colorectal cancer, due to significantly downregulated in colorectal tumors [37]. There has been little research on whether *TMEM236* is involved in growth regulation, only speculating that up-regulation of *TMEM236* may be linked to insulin resistance [38]. As for weight, two missense mutations, rs149243426 and rs10747556, located in the *BTAF1* and *CCDC65* genes, respectively. Past studies have shown that *BTAF1* mutations

can cause post-gastrulation embryonic lethality in mice and results in progressive patchy hair loss from the scalp [39,40]. However, *CCDC65* genes has been associated with a variety of cancers, premature deaths in mice or sperm motility [41,42]. Considering the multiple cases of *BTAF1* and *CCDC65* genes in growth and development, combined with our GWAS results, we believe that the above two genes are likely to play a role in neonatal weight lifting as well. We have identified 109 SNPs suggestively associated with BMI, while no missense mutation SNPs were found. Moving forward, we aim to gather more enrolled BMI data to further enhance the depth of our analysis. After gaining SNP information for the 12 missense mutations, we further analysed the distribution of allele frequencies around the world and within China. We found that the SNP allele frequencies of these 12 missense mutations were consistent across the QD cohort, ChinaMAP and EAS.

Upon further analysis, our findings reveal variations in the frequencies of certain mutations across diverse populations. For instance, rs1042597 was found to be high in the QD cohort, ChinaMAP, and EAS, while rs17868323 and rs17868323 showed lower frequencies, and rs11692021, rs6759892, rs2070959 as well as rs11058978, which also seems to show same trend. This suggests that rs1042597 may serve as a crucial SNP for predicting JAU and JAUR in East Asian populations, and it works the opposite of the rest. It underscores the importance of considering population variability in our future predictions of JAU and JAUR. In contrast, rs148399850, rs927472313, and rs149243426 consistently showed similar patterns across various populations. This implies that these three SNPs likely play comparable roles in determining BW, weight, and height across diverse population groups. The diversity of allele frequencies of rs10747556 across populations suggests that the biomarker is not significantly associated with the populations.

When analyzing within China, two noteworthy points emerge: the first is rs1042597, rs2070959, rs4148323, and rs10747556 exhibit the highest allele frequency in Xizang, but rs6759892, rs2070959, rs4148323 and rs10747556 were almost undistributed in Guangxi Province. Since Xizang is mainly Xizang and Guangxi is mainly Zhuang, we speculate that these results may be caused by ethnic differences. Since 4 of them SNPs are associated with JAU, we have to focus on the prioritize addressing JAU potentially present in newborns from Xizang and Guangxi Province by implementing timely and reliable healthcare programs. Certainly, our hypothesis also requires more demographic evidence on JAU. Whereas rs10747556, a SNP significantly associated with weight, the absence of allele frequencies in the Guangxi population requires further investigation. The second one we found that rs148399850 and rs927472313 seem to show a decrease in allele frequency from northern to southern regions. In particular, rs927472313 was significantly associated with height, and the distribution of height in the Chinese population showed a trend of higher height in the north and lower height in the south [43]. We hypothesized that rs927472313 and its located gene, *TMEM236*, may be a functional gene for height trend in the Chinese population. Furthermore, maybe offers a reference point for elucidating the height variability between northern and southern China at the molecular level. Unfortunately, due to the lack of complete data on Chinese population BW, the function of rs148399850 in Chinese population differences requires more work. In addition to this, we also found rs148399850, rs927472313 and rs149243426 were undistributed in Xizang and Beijing. These three SNPs were significantly associated with BW, height and weight, respectively. Regarding rs1105879, we only observed a lower allele frequency on the southeast coast. The reason for the concordance between Xizang and Beijing in the BW, height and weight traits, and the result of rs1105879 need to be supported by more data in the future.

After aggregating all the candidate genes and conducting functional enrichment analysis, we identified 20 arrays GO (S9 Table), 18 candidate genes and 14 pathways (Fig 4A and S4 Fig). Twenty arrays GO, including the most significant GO, flavonoid glucuronidation, are related to growth and development [44]. Of the 18 candidate genes we found, except for 9 *UGT1A* family genes, 3 genes *PRKCH*, *AHR* and *KCNMA1* have been reported to be involved in neonatal growth and development [45–47]. The remaining 6 candidate genes, we believe, are also important potential targets for neonatal health research. Fourteen significant pathways, such as hsa00053, hsa00140, and hsa05207, these results further corroborate the involvement of ascorbate in the biological mechanisms of JAU, and steroids are used for treatment of hepatic JAU and JAUR [48,49], suggesting a potential link to cancer, along with multiple other biological pathways

previously underexplored in the context of JAU and JAUR. Regarding hsa00040, hsa04270, hsa04390, and hsa05207, since all these pathways are importantly linked to growth and development, especially hsa04390 plays an important role in early neonatal development [50], which implies that our study can effectively explain the molecular landscape of neonatal growth and development. To ensure the coverage of these candidate genes, we displayed the gene distributions and found that they were distributed on all 21 chromosomes except chromosome 15, suggesting that the molecular markers in our NHGI framework can be a relatively comprehensive reference for evaluating neonatal health.

Our study also highlights the causal effects of clinical factors on indicators of HNGI, employing genetic correlation analysis to elucidate the genetic basis of observed clinical impacts. We found positive correlation between HC and BW, and also noted negative correlation between BW and SCS, a rare condition with heterogeneous etiologies [27], is rarely comprehensively analyzed with BW, exemplified by a rare case in an Extremely Low Birth Weight (ELBW) newborn [51]. Another study indicated that, for one group of children, the size of the lumbar vertebral canal was reduced by low BW [52], which aligns with our findings. It is unclear whether this is a rare case or a common one, and more clinical data needs to be collected to further validate it. Despite this case of HC contradicting our findings, the results of genetic correlation of BW still indicate that BW serves as a significant risk indicator for HC and SCS. Newborns with BW abnormalities should be alerted to the risks of HC and SCS, guiding them towards a healthier life. We also focused on the correlation between BW and HDLC, Ca, and MCHC. Multiple previous reports have confirmed their correlation [53–55], which reinforces the reliability of our correlation model, as well as providing a reference for safeguarding the health of newborns with BW. Regarding JAU and JAUR, we found that JAUR was significantly associated with CHF, which is consistent with previous findings [56]. Cir is negatively correlated with JAUR, as liver performance plays a decisive role in JAUR. Regarding the correlation of ARF with JAU and weight, previous studies have shown that ARF causes dramatic weight loss [57] and that ARF is often accompanied by JAU [58]. The correlation between CRF and BMI has been reported previously [59]. As for the correlation of Car with JAUR, JAU, weight, and BMI, previous reports confirmed that Car was associated with weight and BMI [60], but there was no direct evidence to verify the correlation of Car with JAR and JAUR. This finding, suggesting a link between neonatal JAU and Car, provides a molecular basis for predicting the onset of Car. The NS: a complication of massive obesity [61], but no correlation of NS with JAU and JAUR has been reported, and we speculate that renal performance is likely to be an important cause of neonatal JAU.

The implications of our research are vast, offering novel insights into the genetic factors that influence neonatal health and growth. As we integrate these findings with existing knowledge, it becomes increasingly clear that a multidisciplinary approach, combining genetics and clinical medicine, is essential for advancing neonatal care. Future research should focus on longitudinal studies to track the long-term impacts of these genetic factors, as well as on expanding the diversity of populations studied to ensure the global applicability of these findings. Further functional studies and replication in additional cohorts are needed to clarify the biological relevance of these variants. It should also be noted that our cohort was derived from a single region, and therefore caution is needed when generalizing these findings to other populations Future studies should also consider additional confounding factors, such as maternal health conditions, nutrition, and socioeconomic status, to further refine the associations observed. For molecular markers, such as the *ATP7* gene or the 7 novel missense mutant SNPs identified in our study, we believe that full molecular biology validation should be given.

In conclusion, our study has a positive effect on early warning of neonatal JAU and adds to the growing body of evidence that genetic and clinical factors play a significant role in determining neonatal health outcomes. By identifying novel genetic markers, pathways and clinical factors associated with newborn growth and development, we pave the way for more targeted and effective interventions in neonatal care. Continued exploration of the genetic basis of neonatal health, incorporating the insights gained from our research, holds the promise of significantly improving outcomes for newborns not just in China, but worldwide.

## Conclusion

This study presents a comprehensive GWAS of the Qingdao newborns cohort, delineating the genetic architecture underlying HNGI. We identified 778 SNPs and 120 genes significantly associated with HNGI, including 12 missense mutations, of which 7 are novel, and population-specific genetic determinants for neonatal jaundice and early growth patterns, expanding the molecular database for early health assessment.

Beyond variant discovery, our integrative analysis—encompassing allele frequency distributions across populations, functional enrichment of candidate genes, and genetic correlations with clinical risk factors—provides a multi-dimensional perspective on neonatal health. The findings not only enhance our understanding of the biological mechanisms governing early development but also pave the way for personalized neonatal care strategies, potentially enabling early risk prediction and timely intervention for conditions like jaundice.

Future studies in larger, diverse cohorts are warranted to validate these associations and explore the underlying molecular pathways. Ultimately, this work contributes a valuable resource for advancing precision medicine in the critical newborn period.

## Supporting information

**S1 Fig. QQ-plot for JAU, JAUR, BW, Height, Weight and BMI.**
(EPS)

**S2 Fig. Locus zoom plot for loci reaching GWAS statistical significance ($P<1\times10^{-5}$) for JAU, JAUR, BW, Height, Weight and BMI.** The most significant SNPs in each region were plotted in purple.
(EPS)

**S3 Fig. Comparison of birth traits among genotypes of missense variants ($P<1\times10^{-5}$).**
(EPS)

**S4 Fig. GO and KEGG pathway enrichment analysis of GWAS candidate genes for JAU, JAUR, BW, Height, Weight and BMI.** $P<.05$ was considered to be statistically significant.
(EPS)

**S5 Fig. Sensitivity analysis results of SNPs based on the leave-one-out method.**
(EPS)

**S1 Table. The significant single variants associated with JAU.**
(XLSX)

**S2 Table. The significant single variants associated with JAUR.**
(XLSX)

**S3 Table. The significant single variants associated with Birth weight (BW).**
(XLSX)

**S4 Table. The significant single variants associated with Height.**
(XLSX)

**S5 Table. The significant single variants associated with Weight.**
(XLSX)

**S6 Table. The significant single variants associated with BMI.**
(XLSX)

**S7 Table. Allele frequence of significant missense variants (P<1×10⁻⁵) between our cohort and other populations.**
(XLSX)

**S8 Table. Allele frequency of missense variants (P<1×10⁻⁵) per Chinese administrative division based on the Chinese Millionome Database (CMDB).**
(XLSX)

**S9 Table. GO enrichment analysis of all candidate genes for JAU, JAUR, BW, Height, Weight and BMI.**
(XLSX)

**S10 Table. Pathway enrichment analysis of all candidate genes for JAU, JAUR, BW, Height, Weight and BMI.**
(XLSX)

**S11 Table. Genetic correlation between birth traits and 54 clinical factors in BBJ.**
(XLSX)

**S12 Table. Functional prediction of the novel missense mutations identified in the GWAS.**
(XLSX)

**S1 File. URLs.**
(DOCX)

## Acknowledgments

We thank Qingdao Women and Children's Hospital, Qingdao University for their support in the newborn follow-up for this study.

## Author contributions

**Conceptualization:** Shuo Li, Hui Huang.

**Data curation:** Xu Chen, Peina Du, Zhengyang Yao.

**Formal analysis:** Peina Du, Xiaohong Wang.

**Methodology:** Mengyang Xu, Yushan Huang.

**Project administration:** Ya Gao, Guangyi Fan, Xin Jin, Hui Huang, Silin Pan.

**Resources:** Shuo Li, Mingyan Fang, Silin Pan.

**Software:** Peina Du, Shuo Li, Mengyang Xu, Zhaobin Chu, Yue Zhang.

**Validation:** Zhaobin Chu, Xuejie Huan.

**Writing – original draft:** Xu Chen.

**Writing – review & editing:** Xu Chen.

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
