## [Decision Letter · Decision Letter 0]

12 May 2025

Dear Dr. Chen,

Thank you for submitting your manuscript to PLOS ONE. After careful consideration, we feel that it has merit but does not fully meet PLOS ONE’s publication criteria as it currently stands. Therefore, we invite you to submit a revised version of the manuscript that addresses the points raised during the review process.

We look forward to receiving your revised manuscript.

Kind regards,

Nejat Mahdieh

Academic Editor

PLOS ONE

Journal Requirements:

3. We note that Figure 4 in your submission contain [map/satellite] images which may be copyrighted. All PLOS content is published under the Creative Commons Attribution License (CC BY 4.0), which means that the manuscript, images, and Supporting Information files will be freely available online, and any third party is permitted to access, download, copy, distribute, and use these materials in any way, even commercially, with proper attribution. For these reasons, we cannot publish previously copyrighted maps or satellite images created using proprietary data, such as Google software (Google Maps, Street View, and Earth). For more information, see our copyright guidelines: http://journals.plos.org/plosone/s/licenses-and-copyright.

a. You may seek permission from the original copyright holder of Figure 4 to publish the content specifically under the CC BY 4.0 license.

4. We notice that your supplementary figures are uploaded with the file type 'Figure'. Please amend the file type to 'Supporting Information'. Please ensure that each Supporting Information file has a legend listed in the manuscript after the references list.

5. Please upload a copy of Supporting Information Table S1, S2, S3, S4, S5, S6, S7, S8, S9, S10, S11,  which you refer to in your text on page 6, 7, 8, 9.

Reviewers' comments:

Reviewer's Responses to Questions

**Comments to the Author**

1. Is the manuscript technically sound, and do the data support the conclusions?

Reviewer #1: Yes

Reviewer #2: Yes

2. Has the statistical analysis been performed appropriately and rigorously?

Reviewer #1: Yes

Reviewer #2: Yes

3. Have the authors made all data underlying the findings in their manuscript fully available?

Reviewer #1: Yes

Reviewer #2: Yes

4. Is the manuscript presented in an intelligible fashion and written in standard English?

Reviewer #1: Yes

Reviewer #2: Yes

Reviewer #1: Comments to the Author

Thank you for submitting this paper for review. Please, find my comments below:

1.The wording /English language needs improving throughout.

2.The opening statement, "The health and development of newborns are always a concern," is vague. Please specify the current research status and significance of neonatal jaundice and growth patterns.

3.The Introduction lacks epidemiological or clinical context (e.g., global/Chinese incidence rates, risk factors for neonatal jaundice). Please supplement with relevant data.

4.Exclusion criteria for confounders (e.g., preterm birth, maternal diseases) are not described in the Methods. Please clarify inclusion/exclusion criteria.

5.Seven novel missense mutations (e.g., rs1042597) lack functional predictions (e.g., SIFT/PolyPhen-2 scores) or conservation analyses in the Results. Please provide bioinformatics support.

6.Functional validation (e.g. in vitro studies) or larger cohort replication is recommended if available.

7.Study limitations (e.g., regional bias from Qingdao samples) are not addressed. Please acknowledge generalizability constraints.

8.The Conclusion repeats results but does not highlight core contributions (e.g., "First identification of X novel SNPs in Chinese neonates"). Please succinctly summarize innovations.

9.Please outline future research.

Reviewer #2: I appreciate the opportunity to evaluate this timely study on neonatal growth genetics. The authors present a valuable GWAS dataset (n=6,685) with robust quality control, identifying both novel SNPs (e.g., rs148399850) and clinically relevant pathways (e.g., flavonoid glucuronidation). While the rigorous methodology and comprehensive functional enrichment analyses are clear strengths, I recommend clarifying several methodological points and contextualizing findings more deeply to enhance impact.

1.Methods

Please:

• Clarify if recruitment was random; address potential selection bias. Mention inclusion and exclusion criteria

• State if multiple testing correction was applied beyond the GWAS threshold (P =1×10-8).

• For SNPs with P =1×10-5add sensitivity analyses to reduce false positives.

2. Results

Please:

• Resolve the contradiction between "no significant SNPs" and later reporting 109 SNPs (P =1×10-5).

• The reported differences in allele frequencies between populations (e.g., Xizang vs. Guangxi) are interesting. A statistical test (e.g., χ²) could further strengthen these observations, though the trends remain biologically plausible.

3. Discussion

Please:

• Propose functional validation (e.g., in vitro assays for novel SNPs like rs927472313 in TMEM236).

• Limitations:

o Confounding factors (maternal nutrition, SES) were not adjusted for.

o Generalizability to non-East Asian populations is unclear.

**Do you want your identity to be public for this peer review?** For information about this choice, including consent withdrawal, please see our Privacy Policy

Reviewer #1: No

Reviewer #2: **Yes: ** Farnoosh Emami

---

## [Author Response · Author response to Decision Letter 1]

10 Nov 2025

Reply to Reviewer Comments for Manuscript PONE-D-25-10541

“Genetic and Clinical Determinants of Neonatal Jaundice and Growth Patterns in the Qingdao Birth Cohort: A Genome-Wide Association Study”

Editor

Responses:

Thank you for your feedback. We have revised the formatting of our manuscript based on the reference material you provided.

Responses:

We have removed the ethical statement from the Declarations section.

3. We note that Figure 4 in your submission contain [map/satellite] images which may be copyrighted. All PLOS content is published under the Creative Commons Attribution License (CC BY 4.0), which means that the manuscript, images, and Supporting Information files will be freely available online, and any third party is permitted to access, download, copy, distribute, and use these materials in any way, even commercially, with proper attribution. For these reasons, we cannot publish previously copyrighted maps or satellite images created using proprietary data, such as Google software (Google Maps, Street View, and Earth). For more information, see our copyright guidelines: http://journals.plos.org/plosone/s/licenses-and-copyright.

a. You may seek permission from the original copyright holder of Figure 4 to publish the content specifically under the CC BY 4.0 license.

Responses:

We have removed the map and replaced it with a new figure to present the same analysis results.

4. We notice that your supplementary figures are uploaded with the file type 'Figure'. Please amend the file type to 'Supporting Information'. Please ensure that each Supporting Information file has a legend listed in the manuscript after the references list.

Responses:

We have added legends for the supplementary figures and supplementary tables after the references.

5. Please upload a copy of Supporting Information Table S1, S2, S3, S4, S5, S6, S7, S8, S9, S10, S11, which you refer to in your text on page 6, 7, 8, 9.

Responses:

We have uploaded the supplementary table.

Reviewer #1: 

Thank you for submitting this paper for review. Please, find my comments below:

1. The wording /English language needs improving throughout.

Responses:

Thank you for your feedback. We have thoroughly revised the manuscript to improve its linguistic fluency throughout.

2. The opening statement, "The health and development of newborns are always a concern," is vague. Please specify the current research status and significance of neonatal jaundice and growth patterns.

Responses:

Thank you for your feedback. We revised the Background sentence in the abstract to more clearly reflect the clinical relevance and current research context.

3. The Introduction lacks epidemiological or clinical context (e.g., global/Chinese incidence rates, risk factors for neonatal jaundice). Please supplement with relevant data.

Responses:

Thank you for your feedback. We added a brief epidemiological and clinical context to the Introduction to clarify the frequency, clinical relevance, and major risk factors of neonatal jaundice.

4. Exclusion criteria for confounders (e.g., preterm birth, maternal diseases) are not described in the Methods. Please clarify inclusion/exclusion criteria.

Responses:

Thank you for your feedback. We have now clarified the exclusion of preterm cases.

Other maternal complications could not be systematically excluded due to incomplete records. These potential confounding factors need to be considered and controlled for in subsequent studies.

5. Seven novel missense mutations (e.g., rs1042597) lack functional predictions (e.g., SIFT/PolyPhen-2 scores) or conservation analyses in the Results. Please provide bioinformatics support.

Responses:

Thank you for your feedback. We have added functional annotation results for the seven novel missense variants based on SIFT and PolyPhen-2 predictions in the Results section and included the full data in a supplementary table (Table S14).

6. Functional validation (e.g. in vitro studies) or larger cohort replication is recommended if available.

Responses:

Thank you for the suggestion. We agree that functional validation and replication in additional cohorts would strengthen the findings. We have now added a description reflecting this point in the Discussion section.

7. Study limitations (e.g., regional bias from Qingdao samples) are not addressed. Please acknowledge generalizability constraints.

Responses:

Thank you for your feedback. We have now acknowledged the limitation of the regional source of the cohort and noted that caution is required when generalizing the findings to other populations.

8. The Conclusion repeats results but does not highlight core contributions (e.g., "First identification of X novel SNPs in Chinese neonates"). Please succinctly summarize innovations.

Responses:

We have revised the Conclusion to reduce repetition of detailed results and to more clearly highlight the main contributions of this study.

9. Please outline future research.

Responses:

We have outlined the directions for future research in the discussion, including the need for functional validation, replication in additional cohorts, and longitudinal follow-up to assess long-term outcomes.

Reviewer #2: 

 I appreciate the opportunity to evaluate this timely study on neonatal growth genetics. The authors present a valuable GWAS dataset (n=6,685) with robust quality control, identifying both novel SNPs (e.g., rs148399850) and clinically relevant pathways (e.g., flavonoid glucuronidation). While the rigorous methodology and comprehensive functional enrichment analyses are clear strengths, I recommend clarifying several methodological points and contextualizing findings more deeply to enhance impact.

1.Methods

Please:

• Clarify if recruitment was random; address potential selection bias. Mention inclusion and exclusion criteria

Responses:

Thank you for the comment. We have clarified in the Methods that newborns were randomly recruited; preterm infants and those who had recent blood transfusions were excluded.

• State if multiple testing correction was applied beyond the GWAS threshold (P =1×10-8).

Responses:

Thank you for the comment. Only the standard GWAS significance threshold (P < 5×10⁻⁸) was applied in this study, which is widely accepted to control the family-wise error rate (FWER). Because of the limited sample size, loci with P < 1×10⁻⁵ were also explored for functional relevance. Additional corrections such as FDR or Bonferroni can be added if required.

• For SNPs with P =1×10-5add sensitivity analyses to reduce false positives.

Responses:

Thank you for your helpful suggestion. We have conducted the leave-one-out sensitivity analysis (Fig S5), and added the description and statement in the revised manuscript.

2. Results

Please:

• Resolve the contradiction between "no significant SNPs" and later reporting 109 SNPs (P =1×10-5).

Responses:

Thank you for pointing this out. We have revised the results to distinguish significant loci from suggestive associations, clarifying the statistical thresholds and removing ambiguity.

• The reported differences in allele frequencies between populations (e.g., Xizang vs. Guangxi) are interesting. A statistical test (e.g., χ²) could further strengthen these observations, though the trends remain biologically plausible.

Responses:

Thank you for your helpful suggestion. We have now performed χ² tests to assess allele-frequency differences among populations. The corresponding results have been added as Table S8, and a brief description was included in the results section .

3. Discussion

Please:

• Propose functional validation (e.g., in vitro assays for novel SNPs like rs927472313 in TMEM236).

Responses:

Thank you for your helpful suggestion. We will conduct functional verification in our future research, and the corresponding paragraphs have been added to the Discussion section of the manuscript .

• Limitations:

o Confounding factors (maternal nutrition, SES) were not adjusted for.

o Generalizability to non-East Asian populations is unclear.

Responses:

Thank you for your insightful comments. Due to the limitations of our sampling design, we could not comprehensively control for all potential confounding factors. The confounders that were excluded have now been specified in the revised manuscript . We also acknowledge the regional limitation of our cohort, which should be addressed in future studies. Corresponding statements regarding confounding factors and regional generalizability have been added to the discussion section.

---

## [Decision Letter · Decision Letter 1]

25 Nov 2025

Genetic and Clinical Determinants of Neonatal Jaundice and Growth Patterns in the Qingdao Birth Cohort: A Genome-Wide Association Study

PONE-D-25-10541R1

Dear Dr. Chen,

We’re pleased to inform you that your manuscript has been judged scientifically suitable for publication and will be formally accepted for publication once it meets all outstanding technical requirements.

Kind regards,

Nejat Mahdieh

Academic Editor

PLOS ONE

Additional Editor Comments (optional):

Reviewers' comments:

Reviewer's Responses to Questions

**Comments to the Author**

Reviewer #2: All comments have been addressed

2. Is the manuscript technically sound, and do the data support the conclusions?

Reviewer #2: Yes

3. Has the statistical analysis been performed appropriately and rigorously?

Reviewer #2: Yes

4. Have the authors made all data underlying the findings in their manuscript fully available?

Reviewer #2: Yes

5. Is the manuscript presented in an intelligible fashion and written in standard English?

Reviewer #2: Yes

Reviewer #2: The authors have carefully responded to all reviewer comments. the revised version demonstrates significant improvement in both structure and content.

**Do you want your identity to be public for this peer review?** For information about this choice, including consent withdrawal, please see our Privacy Policy

Reviewer #2: No

---

## [Editor Report · Acceptance letter]

PONE-D-25-10541R1

PLOS One

Dear Dr. Chen,

I'm pleased to inform you that your manuscript has been deemed suitable for publication in PLOS One. Congratulations! Your manuscript is now being handed over to our production team.

Kind regards,

on behalf of

Dr. Nejat Mahdieh

Academic Editor

PLOS One